# IGF-1 Signaling Modulates Oxidative Metabolism and Stress Resistance in ARPE-19 Cells Through PKM2 Function

**DOI:** 10.3390/ijms252413640

**Published:** 2024-12-20

**Authors:** Silvia Ravera, Alessandra Puddu, Nadia Bertola, Daniela Verzola, Elisa Russo, Davide Maggi, Isabella Panfoli

**Affiliations:** 1Department of Experimental Medicine, University of Genoa, Via De Toni 14, 16132 Genova, Italy; silvia.ravera@unige.it; 2IRCCS Ospedale Policlinico San Martino, Largo Rosanna Benzi 10, 16132 Genova, Italy; nadia.bertola@hsanmartino.it; 3Department of Internal Medicine and Medical Specialties, University of Genova, Viale Benedetto XV 6, 16132 Genova, Italy; daniela.verzola@unige.it (D.V.); elisa.russo@unige.it (E.R.); davide.maggi@unige.it (D.M.); 4Unit of Nephrology, Dialysis and Transplantation, IRCCS Ospedale Policlinico San Martino, Largo Rosanna Benzi 10, 16132 Genova, Italy; 5Department of Pharmacy—(DIFAR), University of Genoa, Viale Benedetto XV 3, 16132 Genova, Italy; panfoli@difar.unige.it

**Keywords:** retinal pigment epithelium, Klotho, insulin-like growth factor 1, isoform M2 of pyruvate kinase, energy metabolism, redox balance

## Abstract

The retinal pigment epithelium (RPE) contributes to retinal homeostasis, and its metabolic dysfunction is implied in the development of retinal degenerative disease. The isoform M2 of pyruvate kinase (PKM2) is a key factor in cell metabolism, and its function may be affected by insulin-like growth factor 1 (IGF-1). This study aims to investigate the effect of IGF-1 on PKM2 modulation of RPE cells and whether co-treatment with klotho may preserve it. ARPE-19 cells, an ex vivo model of human pigmented epithelium, were exposed to IGF-1. Then, we evaluated PKM2 expression, dimerization and subcellular localization, energy metabolism, and redox balance, and whether pre-treatment with Klotho may antagonize the effects of IGF-1. The results show that IGF-1 favors PKM2 dimerization, thus reducing the activity of PKM2 and leading to an altered cellular energy status coupled with reduced oxidative stress. In conclusion, PKM2 plays a pivotal role in the modulation of RPE metabolism and redox balance and could explain the mechanisms through which IGF-1 participates in the pathogenesis of some retinal diseases. Klotho may exert protective effects by mitigating the IGF-1 signal and its effect on mitochondrial function.

## 1. Introduction

The retinal pigment epithelium (RPE) is a monolayer of polarized cells located between the choroid and retina that functions as a blood–retinal barrier, contributing to retinal homeostasis [1]. The importance of the RPE in the retina is also related to its metabolic contribution to retinal function maintenance. Indeed, RPE metabolic disorders may play a pivotal role in aging and retinal degenerative diseases [2,3,4]. In particular, it has been found that the expression of glycolytic genes increases in RPE cells in degenerative diseases [4,5], and that altered mitochondrial metabolism is the primary cause of RPE dysfunction [2,6]. Moreover, photoreceptors and the retinal pigment epithelium (RPE) are engaged in crosstalk, and mutually influence each other [7,8]. In the past, it has been proposed that the RPE provides glucose to photoreceptors by recycling the lactate they produce [9], thereby sustaining their energy requirements. More recently, our study suggests that under pathological conditions oxidative stress produced by photoreceptors may negatively impact the phagocytic capacity of the RPE, exacerbating oxidative damage and triggering a vicious cycle [10]. In any case, the disrupted balance between photoreceptors and the RPE is consistently associated with dysfunction in oxidative phosphorylation (OxPhos) [2,6] and an increase in glycolytic enzymes [4,5,11] as an attempt to restore cellular energy status. Within this framework, our focus has shifted to isoform M2 of pyruvate kinase (PKM2), a glycolytic enzyme involved in the modulation of the glucose catabolic pathway [12]. PKM2 exists in tetrameric and dimeric form, both composed of the same monomer, but with different biological actions [13]: tetrameric PKM2 functions as a cytosolic enzyme, catalyzing the final step of glycolysis by converting phosphoenolpyruvate into pyruvate and producing ATP [14]. Conversely, dimeric PKM2 translocates into the nucleus to regulate gene expression. The interconversion between these forms can significantly influence tumor cell energy supply, epithelial–mesenchymal transition (EMT), invasion and metastasis, and cell proliferation [13,15]. Although the function of PKM2 has been mainly investigated in cancer, it also plays a significant role in the retina. PKM2 loss leads to decreased rod function [16], and its expression is reduced in the retina of db/db mice (a type 2 diabetes animal model) [17]. Conversely, increased PKM2 expression has been observed in RPE cells with upregulated glycolytic capacity, such as when cultured under high-glucose conditions [18] or induced to undergo EMT [19]. Growth factor signaling may affect the catalytic activity of PKM2, inducing its nuclear translocation, as we previously demonstrated in treating a human lung adenocarcinoma cell line with insulin-like growth factor 1 (IGF-1) [20]. Moreover, we have found that the increased nuclear localization of PKM2 is associated with enhanced expression of HIF1α, HK2, and GLUT1, leading to glucose entrapment and decreased cellular PK activity [20]. On the other hand, the activation of IGF-1 signaling has been implied in the pathogenesis of ocular diseases characterized by neovascularization, such as age-related macular degeneration (AMD) and proliferative diabetic retinopathy (PDR), principally due to IGF-1-stimulated vascular endothelial growth factor A (VEGF-A) secretion [21]. This effect is mediated by the HIF1α increased expression in RPE and may be reduced by the downregulation of caveolin-1 expression [22] and by the treatment with the protein Klotho, a protein with antiaging proprieties and involved in retinal function maintenance [23,24,25,26,27,28]. These findings, together with the evidence that IGF-1 downregulates PK activity, suggest that IGF1 receptor (IGF1R) activation may be the link between metabolic changes and RPE dysfunction. In addition, the metabolic changes induced by IGF-1 appear to be modulated by Klotho.

Therefore, this study aims to investigate the effect of PKM2 modulation induced by IGF-1 on the energy metabolism and redox balance of ARPE-19 cells, an RPE model, and whether treatment with soluble Klotho may preserve it.

## 2. Results

### 2.1. PKM2 Expression

Investigating whether IGF-1 may affect the expression of PKM2 and its configuration into a dimer, our data show that treatment with IGF-1 significantly increased the expression of total PKM2 (Figure 1A) in ARPE-19 cells, inducing a time-dependent increase in the dimeric form (Figure 1B).

### 2.2. Cellular Localization of PKM2

Furthermore, due to its transition from tetrameric to dimeric form, increased translocation of PKM2 into the nucleus was observed following IGF-1 treatment. Specifically, after 4 h, PKM2 is mainly localized in the perinuclear zone, while after 24 h, it is also present in the nucleus (Figure 2).

### 2.3. PKM2 Activity

As expected, the conformational change in PKM2 also affected the enzymatic activity involved in phosphoenolpyruvate conversion to pyruvate. The data shown in Figure 3 demonstrate that, as early as two hours after treatment, catalytic activity decreased by approximately 85%, reaching its minimum at 4 h post IGF-1 addition. After 24 h of treatment, PK activity partially recovered. Interestingly, the addition of Klotho to the IGF-1 treatment completely inhibits its effect.

### 2.4. Mitochondrial Energy Metabolism

Since PK catalytic activity plays a pivotal role in mitochondrial energy metabolism, with pyruvate being a principal substrate for the Krebs cycle, we assayed aerobic ATP synthesis and oxygen consumption rate (OCR) to investigate the effect of PKM2 dimerization on OxPhos. As depicted in Figure 4A,B, ATP synthesis and OCR decreased by 44 and 40%, respectively, 4 h after the IGF-1 treatment, suggesting that the reduction in PK activity may affect the pyruvate pool required for mitochondrial energy production. However, OxPhos inhibition partially recovered 24 h after treatment, as ATP synthesis and OCR increased by approximately 34% compared to their levels at 4 h, suggesting a restoration of aerobic mitochondrial metabolism. The analysis of metabolic dependence, reported in Figure 4C,D, reveals that after 24 h of IGF-1 treatment, ARPE-19 cells preferentially used glutamine and fatty acids (+190% and +260%, respectively, compared to the control, for both ATP synthesis and OCR) as energy substrates, rather than pyruvate (−70% compared to the untreated sample), suggesting that the cells activated an adaptive response to the reduced pyruvate availability associated with decreased PK activity. Similarly to PK activity, the addition of Klotho prevented the effects of IGF-1 on OxPhos metabolism, both in terms of activity and substrate dependency.

Usually, a reduction in aerobic metabolism induces an increase in lactic fermentation to compensate for the decreased ATP production and to convert NADH to NAD^+^. Therefore, in ARPE-19 cells, lactate dehydrogenase (LDH) activity and the glycolytic rate were evaluated. The data show that LDH activity in ARPE-19 cells treated with IGF-1 follows a similar pattern to that of PK, decreasing 2 h after treatment, reaching its lowest value at 4 h, and then partially recovering 24 h post IGF-1 addition (Figure 5A). This modulation of LDH has not been observed in cells treated with IGF-1 in the presence of Klotho. The reduction in LDH activity is associated with a decrease in the anaerobic glycolytic rate compared to the cells treated with IGF-1 + Klotho, suggesting a metabolic block at both anaerobic and aerobic levels (Figure 5B).

### 2.5. Cellular Energy Status of ARPE-19 Cells Treated with IGF-1

Since OxPhos and glycolysis are the principal sources of chemical energy production, the cellular energy status was also evaluated. The results show that IGF-1 treatment reduces ATP levels and increases AMP concentration, following the same pattern as PK and LDH activity (Figure 6A and Figure 6B, respectively), resulting in a significant decrease in the ATP/AMP ratio (Figure 6C), an indicator of cellular energy status. These effects are completely reversed by the Klotho addition.

### 2.6. Evaluation of Oxidative Stress

Given that OxPhos is one of the principal sources of oxidative stress, the impact of IGF-1 on oxidative damage accumulation was investigated. As shown in Figure 6, treatment with IGF-1, but not with IGF-1 + Klotho, reduces the malondialdehyde (MDA) accumulation (Figure 7A), a marker of lipid peroxidation, after 4 h from the treatment, suggesting that the reduction in MDA is consequent to the OxPhos reduction. Additionally, treatment with IGF-1 also increases, with the same temporal trend of MDA, the activity of glutathione reductase (GR) and glutathione peroxidase (GPx) (Figure 7C and Figure 7D, respectively), two enzymes involved in antioxidant defenses, and of glucose-6-phosphate dehydrogenase (G6PD), the first enzyme of the cytosolic pentose phosphate pathway (PPP) involved both in redox balance and building block formation.

### 2.7. Activation of IGF1R

To investigate the mechanisms through which Klotho counteracts IGF-1 action, IGF1R activation was evaluated. The results show that IGF-1 treatment significantly increases the phosphorylation of IGF1R, IRS-1, and AKT in ARPE-19 cells. In detail, the phosphorylation of IGF1R decreases after 20 min of IGF-1 exposure (Figure 8A,C), whereas AKT and IRS-1 phosphorylation is still sustained (Figure 8B). Conversely, Klotho reduces IGF1R phosphorylation and its intracellular downstream signaling.

### 2.8. Subcellular Localization of PKM2

To verify whether the reduced activation of IGF-1 signaling affects the intracellular localization of PKM2, we analyzed the subcellular localization of PKM2 in cells treated with IGF-1 and Klotho. We found that IGF-1 does not affect the amount of PKM2 in the cytosolic fraction (F1) but induces a time-dependent localization of PKM2 firstly in the soluble nuclear and then in the chromatin-bound fractions (F3, F4, Figure 9B,C). Concomitant treatment with Klotho retains PKM2 in the soluble fraction of the nucleus (F3, Figure 9B) and reduces the PKM2 amount in the chromatin-bound fraction (F4, Figure 9C).

## 3. Discussion

The retina is one of the most metabolically active tissues in the body and is considered a metabolic ecosystem in which the dysfunction of RPE cells may compromise the function and viability of the other retinal cells [10,29]. In this context, PKM2 is a crucial factor in maintaining metabolic retinal homeostasis and may be a crossroad between metabolic changes and retinal diseases. However, the role of PKM2 in retinal metabolism has been primarily described in photoreceptors [30,31,32,33,34]. Therefore, herein, the role of PKM2 in RPE was evaluated in ARPE-19 cells treated with IGF-1.

The data show that IGF-1 reduces the activity of PKM2 in ARPE-19 cells, consistent with our previous observations in a human lung adenocarcinoma cell line, suggesting that activation of IGF1R signaling is associated with a decreased activity of the pyruvate kinase enzyme across multiple cell types. Interestingly, the reduced enzymatic activity of PKM2 occurs despite IGF-1 stimulating PKM2 expression. Since the PKM2 activity depends on its three-dimensional conformation [13], with optimal activity in its homotetrameric form, this suggests that IGF-1 favors the formation of the dimeric rather than the tetrameric form of PKM2 or leads to the formation of less stable tetramers. Consistently, the ratio between the dimeric and the tetrameric form of PKM2 was increased in ARPE-19 cells treated with IGF-1. Considering that the dimeric state of PKM2 corresponds to a lower PK activity compared to the tetrameric state [13], the increased expression of PKM2 during IGF-1 exposure may represent a compensatory response to decreased PK activity. In addition, Koo et al. demonstrated that PKM2 positively regulates IGF1R expression [35], suggesting the existence of a positive feedback loop that sustains IGF-1 signaling to maintain PKM2 in dimeric form. It is well known that dimeric PKM2 can translocate to the nucleus and act as a transcriptional factor, promoting the expression of genes involved in glycolysis and proliferation [13,36]. Images of ARPE-19 cells treated with IGF-1 reveal a time-dependent translocation of PKM2 in the nucleus. In addition, Western blot analyses, besides confirming the nuclear translocation of PKM2, demonstrated that PKM2 establishes a stable binding to chromatin in ARPE-19 cells, suggesting that it may work as a mediator between IGF-1 intracellular signaling and changes in gene expression. The activation of IGF-1 signaling in RPE cells has been associated with the pathogenesis of ocular diseases characterized by neovascularization, such as AMD and PDR [37], principally because IGF-1 stimulates VEGF-A secretion [21]. Therefore, it can be speculated that the expression of VEGF-A induced by IGF-1 in RPE cells [22,38] may be mediated by the transcriptional activity of PKM2, as occurs in cancer cells [13,15,39,40]. Additionally, the reduction in PK catalytic activity affects the energy metabolism of ARPE-19 cells. Specifically, IGF-1 addition causes a decrease in oxidative phosphorylation, both in terms of oxygen consumption and ATP synthesis, within the first 4 h of treatment. These data can be explained by the fact that lower PK activity leads to reduced production of pyruvate, a primary substrate for aerobic metabolism. However, after 24 h from IGF-1 treatment, ARPE-19 cells partially recover OxPhos functionality, albeit with a shift in their reliance on energy substrates, moving from a preference for pyruvate to a greater dependence on glutamine and fatty acids. Nonetheless, as shown in Appendix A, glucose consumption is higher in IGF-1-treated cells compared to those incubated with IGF-1 + Klotho, suggesting that glucose is utilized in other metabolic pathways. A likely hypothesis is that glucose is used in the PPP to provide building blocks for cellular proliferation and reduced coenzymes (NADPH) to enhance antioxidant defenses [41], as suggested by the G6PD activity increment. On the other hand, it is well known that the lower activity of PKM2 stimulates the PPP, promoting the synthesis of macromolecules and inhibiting the production of ROS [42,43]. These data suggest that IGF-1 in the retina could help to reduce oxidative stress production [37], slowing down OxPhos and related ROS production, as well as increasing antioxidant capacity. On the other hand, it has already been demonstrated in other cell models that treatment with IGF-1 induces an increase in GPx activity [44] and glutathione synthesis [45]. However, ARPE-19 cells cannot compensate for the reduction in aerobic metabolism by incrementing the anaerobic glycolysis due to decreased PK activity, resulting in an altered cellular energy status, as evidenced by the ATP/AMP ratio. Thus, from a metabolic perspective, IGF-1 induces a shift in energy metabolism via PKM2, making the cell more capable of proliferation but less able to respond to changes in homeostasis due to the low availability of ATP. In other words, IGF-1 appears to have a Janus-faced effect on the RPE: it improves the redox state, decreasing the oxidative damage accumulation, but contemporarily reduces the energy availability and modulates gene expression in favor of inflammation and neurodegeneration. On the other hand, this dual IGF-1 behavior has already been described as it displays both a pro- and anti-inflammatory effect [37]. Interestingly, the presence of Klotho restores PK activity to the levels of untreated cells. In detail, Klotho reduces the intracellular IGF-1 signaling, decreasing the phosphorylation of both its receptor and principal substrates IRS-1 and AKT. These results agree with previous evidence showing that AKT mediates the effects of IGF-1 on PKM2 function [20]. Moreover, Klotho decreases the ability of PKM2 to bind to chromatin. Consequently, it may prevent the induction of transcription of target genes that support glycolysis, thus preserving OxPhos. On the other hand, it is known that Klotho is an anti-aging gene expressed in many tissues, including the retina. Evidence shows that Klotho regulates multiple mechanisms critical for maintaining the homeostasis and functions of retinal cells, and its depletion results in the gradual deterioration of retinal function [23], suggesting it is a potential therapeutic target for retinal degenerative diseases [26]. Specifically, Klotho is a pivotal regulator of the homeostasis and functions of RPE cells by protecting against oxidative stress, inducing expression of MERTK/AXL/TYRO3 to enhance phagocytosis, inhibiting IGF-1-induced VEGF-A secretion, regulating pigment synthesis, upregulating proteins related to mitochondrial activity, and preventing senescence-like morphological changes and EMT induced by TGF-β2 [23,24,28].

In conclusion, the data reported herein show that PKM2 modulation induced by IGF-1 signaling could be responsible for RPE dysfunction and, consequently, for the impairment of the retinal ecosystem. These dynamics confirm that PKM2 plays a pivotal role in the modulation of RPE metabolism and redox balance and could explain the mechanisms through which IGF-1 participates in the pathogenesis of retinal diseases. At the same time, the data indicate that the protective effect of Klotho could be due to its ability to mitigate the IGF-1 signal and its effect on mitochondrial function.

## 4. Materials and Methods

### 4.1. Cell Culture and Experimental Conditions

The human cell line ARPE-19 (American Type Culture Collection, Manassas, VA, USA) from passages 22 to 28 were grown in DMEM/F12 1:1 medium (Life Technologies Italia, Milan, Italy) supplemented with 10% fetal bovine serum and 2 mmol/L glutamine (Euroclone, Milan, Italy) at 37 °C in 5% CO_2_. The cell medium was replaced every 2 days. The cells were grown to confluence, removed with trypsin-EDTA (Euroclone, Milan, Italy), and then seeded in multiwell plates for all experiments. Once confluence was reached, the cells were pre-treated for 1 h with 400 pmol/L Klotho (R&D System Inc., Minneapolis, MN, USA) and then exposed for 2, 4, and 24 h to 100 nmol/L IGF-1 (Life Technologies Italia, Milan, Italy) and processed for each analysis.

### 4.2. Cell Immunofluorescence

ARPE-19 cells were seeded and treated on chamber slides as described above. After a five-minute incubation in cold methanol, the cells were exposed to anti-PKM2 (dilution 1:100 in PBS/Tween 0.05%) (D78A4, cat. 4053, Cell Signaling Technology, Beverly, MA, USA) for 30 min at room temperature (RT). Then, secondary antibodies conjugated to Alexa Fluor 594 were used for 30 min at RT (1:400 in PBS) (Life Technologies Italia, Milan, Italy). Finally, the nuclei were counterstained with DAPI and observed under a fluorescence microscope (Leica Microsystems GmbH, Wetzlar, Germany).

### 4.3. PKM2 Crosslinking

To cross-link PKM2, a batch of cells was washed with PBS after treatments and incubated for 30 min at room temperature with 500 µM DSS (Disuccinimidyl suberate, 21655, Thermo Fisher Scientific, Waltham, MA, USA) for the detection of the tetramers, dimers, and monomers of PKM2. Then, the cells were washed with PBS and lysed to be analyzed by Western blot.

### 4.4. Intracellular Signaling

To investigate IGF-I signaling, ARPE-19 cells were serum-starved overnight, then exposed for 10 min to 100 nmol/L recombinant human IGF-1. After treatment, the cells were lysed in RIPA buffer supplemented with protease and phosphatase inhibitor cocktails (all from Pierce Biotechnology, Rockford, IL, USA), and protein concentration was determined using the BCA protein assay Kit (Pierce Biotechnology, Rockford, IL, USA).

### 4.5. Cell Lysis and Subcellular Fractionation

At the end of the experiments, a set of ARPE-19 cells were lysed in RIPA buffer supplemented with protease and phosphatase inhibitors. Another set of cells was processed for subcellular fractionation using the Subcellular Protein Fractionation Kit (Pierce Biotechnology, Rockford, IL, USA) according to the manufacturer’s instructions. Briefly, various cellular compartments were isolated by sequential addition of different extraction buffers to the cell pellet. Each subcellular fraction was collected after centrifugation and stored at −80 °C. Cytosolic, nuclear soluble, and chromatin-bound protein extracts obtained from each experimental condition were used for immunoblot analysis. The protein concentration of each sample was determined using the BCA Protein Assay Kit.

### 4.6. Immunoblot

Fifteen micrograms of total cell lysate or subcellular fractions were separated through denaturing electrophoresis (SDS-PAGE) on 4–20% gradient gels (Life Technologies Italia, Milan, Italy) and transferred onto nitrocellulose using iBlot system (Life Technologies Italia, Milan, Italy). Filters were blocked in Protein Free T20 Blocking Buffer (Pierce Biotechnology, Rockford, IL, USA) and incubated overnight at 4 °C with the following primary specific antibodies: PKM2 (D78A4, cat. 4053), GAPDH (14C10, cat. 2118), phAKT (Ser473, D9E, cat. 4060), phIRS-1 (Tyr895, cat. 3070), phIGF1R (Tyr980, C14A11, cat 4568), H3 (D1H2, cat. 4499), HDAC2 (3F3, cat. 5113), and β-ACT (8H10D10 cat. 3700) from Cell Signaling Technology, Beverly, MA, USA. Secondary specific horseradish peroxidase-linked antibodies were added for 1 h at room temperature. The bound antibodies were detected using the enhanced chemiluminescence lighting system (LiteAblot EXTEND, Euroclone, Milan, Italy), according to the manufacturer’s instructions. Each membrane was stripped (Restore PLUS Western blot Stripping Buffer, Pierce Biotechnology, Rockford, IL, USA) and probed for β-actin or GAPDH (Cell Signaling Technology, Beverly, MA, USA) to verify equal protein loading. Bands of interest were quantified by densitometry using the Alliance 1D software. The results were expressed as a percentage of CTR (defined as 100%).

### 4.7. Pyruvate Kinase Assay

Pyruvate kinase (PK, EC: 2.7.1.40) activity was assayed following the NADH oxidation at 340 nm in a double-beam spectrophotometer (UNICAM UV2, Analytical S.n.c., Borgotaro (PR), Italy) on 20 μg of total protein. The assay mixtures contained: 100 mM Tris HCl pH 8, 10 mM phosphoenolpyruvate, 0.2 mM NADH, and 1 IU/mL of lactate dehydrogenase [20].

### 4.8. OxPhos Activity Evaluation

OxPhos function was evaluated in 2 × 10^5^ ARPE-19 cells permeabilized with 0.01% digitonin and resuspended in the growth medium, measuring aerobic ATP synthesis through FoF1 ATP synthase and the oxygen consumption rate (OCR).

ATP synthesis was triggered by adding 0.1 mM ADP and was monitored employing the luciferin/luciferase chemiluminescence method by a luminometer (GloMax^®^ 20/20 Luminometer, Promega, Milan, Italy) every 30 s for 2 min. ATP standard solutions, ranging from 10^−8^ to 10^−5^ M, were used for calibration (luciferin/luciferase ATP bioluminescence assay kit CLS II, Roche, Basel, Switzerland).

OCR was evaluated by an amperometric electrode (Unisense Microrespiration, Unisense A/S, Aarhus, Denmark) in a closed chamber.

Both in OCR and ATP synthesis evaluation, 3 μM BPTES, 4 μM Etomoxir, or 2 μM UK5099 were used to assess the energy substrates’ dependence. BPTES is a glutaminase inhibitor used to assess metabolic dependence on glutamine; etomoxir is a CTP1-A inhibitor used to assess metabolic dependence on fatty acids, and UK5099 is mitochondrial pyruvate carrier inhibitor used to assess metabolic dependence on pyruvate and consequently on glucose [46].

### 4.9. Lactate Dehydrogenase Activity Assay

Lactate dehydrogenase (LDH; EC 1.1.1.27) activity was measured on 20 μg of total protein, following the NADH oxidation at 340 nm in a double-beam spectrophotometer (UNICAM UV2, Analytical S.n.c., Borgotaro (PR), Italy). The reaction mixtures contained: 100 mM Tris-HCl pH 7.4, 0.2 mM NADH, and 5 mM pyruvate [47].

### 4.10. Anaerobic Glycolysis Yield Evaluation

The anaerobic glycolysis yield was determined by calculating the percentage of real lactate released by ARPE-19 cells in the growth medium relative to the theoretical lactate production, calculated as twice the amount of glucose consumed (since, in exclusive anaerobic metabolism, one glucose molecule is converted into two lactate molecules) [46].

Glucose consumption in the growth medium was assessed by measuring NADP reduction at 340 nm. For this, 10 μL of the growth medium was added to a solution containing 50 mM Tris-HCl pH 8.0, 1 mM NADP, 10 mM MgCl_2_, and 2 mM ATP. Spectrophotometric analysis of the samples was conducted before and after the addition of 4 μg of purified hexokinase (HK) and glucose-6-phosphate dehydrogenase (G6PD).

Lactate concentration in the growth medium was measured spectrophotometrically by monitoring NAD^+^ reduction at 340 nm. The assay medium consisted of 10 μL of growth medium, 100 mM Tris-HCl pH 8.0, and 5 mM NAD^+^. Samples were analyzed spectrophotometrically before and after the addition of 4 μg of purified LDH.

Both sets of data were normalized to the cell number, and the data are reported in Appendix A.

### 4.11. Intracellular ATP and AMP Concentration Evaluation

The quantification of ATP and AMP was performed using the enzyme coupling method. For both assays, 20 μg of total protein was utilized. ATP was measured by monitoring NADP reduction at 340 nm. The reaction medium consisted of 100 mM Tris-HCl pH 8.0, 1 mM NADP, 10 mM MgCl_2_, and 5 mM glucose, in a final volume of 1 mL. Spectrophotometric analysis of the samples was conducted before and after the addition of 4 μg of purified hexokinase along with glucose-6-phosphate dehydrogenase. AMP was measured by observing NADH oxidation at 340 nm. The reaction medium included 100 mM Tris-HCl pH 8.0, 75 mM KCl, 5 mM MgCl_2_, 0.2 mM ATP, 0.5 mM phosphoenolpyruvate, 0.2 mM NADH, 10 IU adenylate kinase, 25 IU pyruvate kinase, and 15 IU lactate dehydrogenase [47].

### 4.12. Lipoperoxidation Evaluation

To assess oxidative damage, the concentration of MDA, a lipid peroxidation marker, was measured using the thiobarbituric acid reactive substance (TBARS) assay. The TBARS reagent consisted of 26 mM TBA and 15% trichloroacetic acid in 0.25 N HCl. A total of 600 μL of the TBARS solution was mixed with 50 μg of total protein dissolved in 300 μL of Milli-Q water. The mixture was incubated at 95 °C for 60 min. Afterward, the samples were centrifuged at 20,000g for 2 min, and the supernatants were analyzed spectrophotometrically at 532 nm [46].

### 4.13. Antioxidant Enzyme Activity Evaluation

G6PD activity was assayed spectrophotometrically following NADP reduction at 340 nm, with a solution containing 100 mM Tris-HCl (pH 7.4), 0.5 mM NADP, and 10 mM glucose-6-phosphate [10].

GR activity was assayed following the oxidation of NADPH with a spectrophotometric analysis at 340 nm. The assay solution contained 100 mM Tris-HCl (pH 7.4), 1 mM EDTA, 5 mM GSSH, and 0.2 mM NADPH [10].

GPx activity was assayed following the decomposition of H_2_O_2_ at 240 nm, using an assay solution containing 100 mM Tris-HCl (pH 7.4), 5 mM H_2_O_2_, and 5 mM GSH. Since H_2_O_2_ is also a substrate of catalases, GPx activity was obtained by subtracting the result of this assay from the catalase activity values (catalase was assayed spectrophotometrically following the H_2_O_2_ decomposition at 240 nm) [10].

### 4.14. Statistical Analysis

The results reflect a minimum of three separate experiments. Data analysis was performed with GraphPad Prism 10 software (GraphPad Software, San Diego, CA, USA). The values are presented as mean ± SD, and an unpaired *t*-test or a one-way ANOVA followed by Tukey’s multiple comparison test were used for statistical analysis. Differences were considered statistically significant when *p* < 0.05.

## Figures and Tables

**Figure 1 ijms-25-13640-f001:**
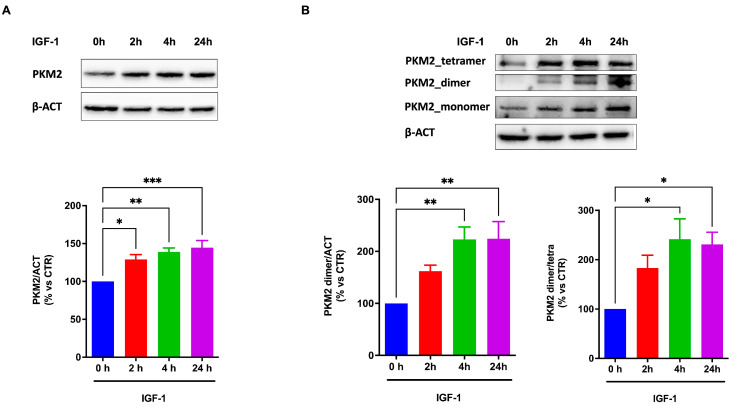
IGF-1 increases the expression and dimerization of the isoform M2 of pyruvate kinase (PKM2) in ARPE-19. (**A**) Protein expression levels of PKM2 in ARPE-19 cells treated with IGF-1 for 2, 4, and 24 h. (**B**) Analysis of the different structural forms of PKM2 in ARPE-19 cells treated with IGF-1 for 2, 4, and 24 h. Data are representative of three independent experiments (n = 3) and are indicated as means ± SD. Statistical analysis was performed with one-way ANOVA followed by Tukey’s multiple comparisons test. *, **, and *** indicate *p* < 0.05, 0.01, or 0.0001, respectively.

**Figure 2 ijms-25-13640-f002:**
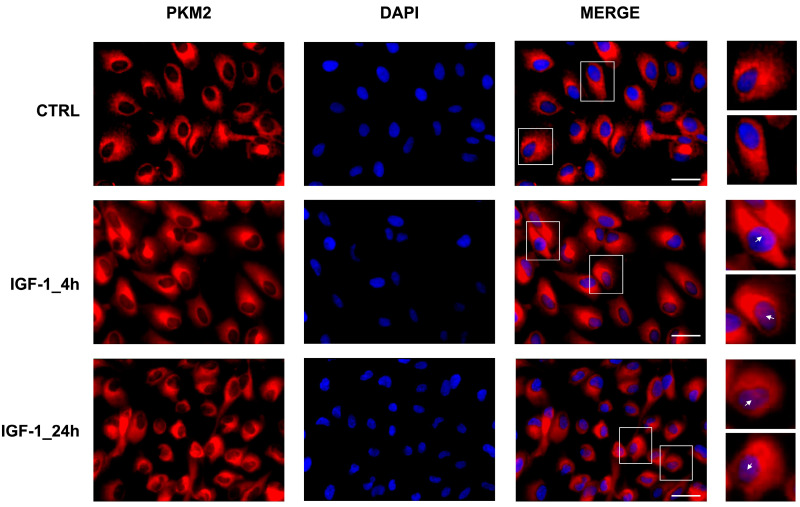
Representative immunofluorescence images of the M2 isoform of pyruvate kinase (PKM2) in ARPE-19 cells treated with IGF-1 for 4 and 24 h (n = 3). The red signal represents the immunolocalization of PKM2 during IGF-1 treatment, while cell nuclei are stained blue with DAPI. White scale bars correspond to 15 μm. The higher-magnification images correspond to the areas enclosed by white squares for each condition in the merged-signals column. The white arrows indicate the nuclear localization of PKM2.

**Figure 3 ijms-25-13640-f003:**
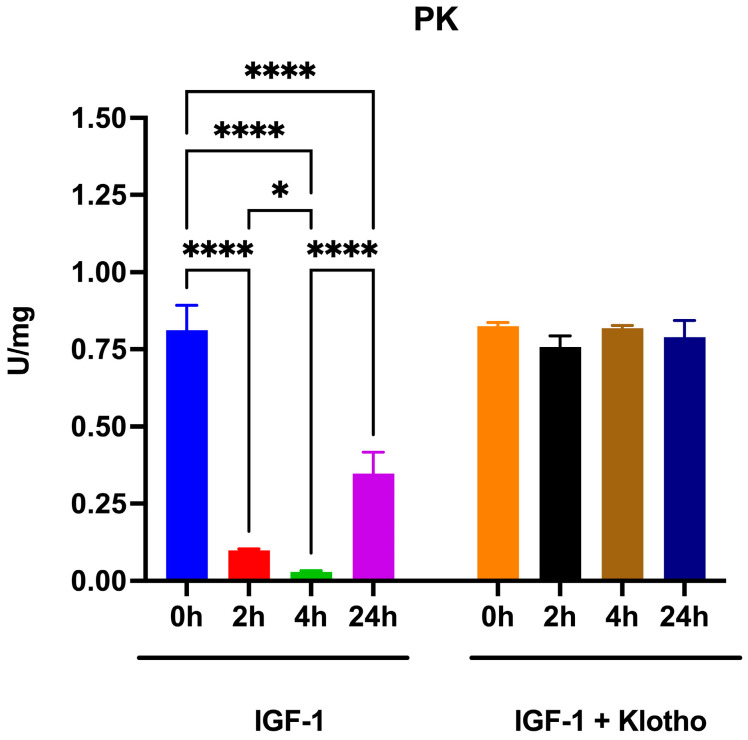
Pyruvate kinase (PK) activity in ARPE-19 cells treated with IGF-1 in the absence or presence of Klotho. The graph shows the PK activity in ARPE-19 cells treated with IGF-1 for 2, 4, and 24 h in the absence or presence of Klotho. Data are representative of four independent experiments (n = 4) and are indicated as means ± SD. Statistical analysis was performed with one-way ANOVA followed by Tukey’s multiple comparisons test. * and **** indicate *p* < 0.05 or 0.0001, respectively.

**Figure 4 ijms-25-13640-f004:**
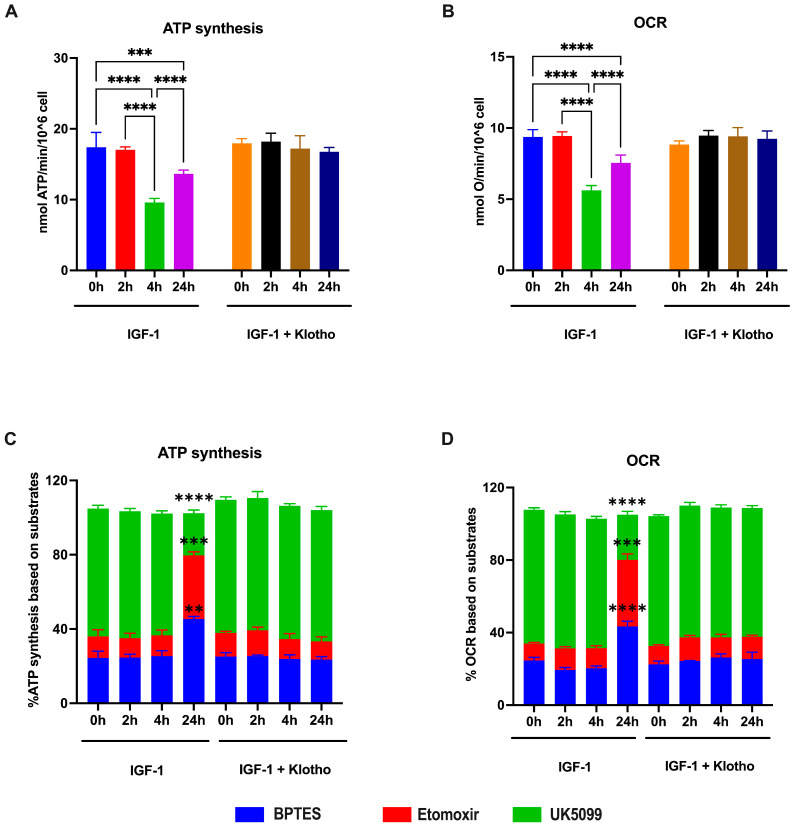
Evaluation of ATP synthesis and oxygen consumption rate (OCR) as markers of OxPhos activity in ARPE-19 cells treated with IGF-1 in the absence or presence of Klotho. All data were obtained from ARPE-19 cells treated with IGF-1 for 2, 4, and 24 h in the absence or presence of Klotho. (**A**) ATP synthesis through F_o_F_1_ ATP synthase; (**B**) OCR; percentages of ATP synthesis (**C**) and OCR (**D**) sensitive to BPTES (a glutaminase inhibitor used to assess metabolic dependence on glutamine), Etomoxir (a CTP1-A inhibitor used to assess metabolic dependence on fatty acid), and UK5099 (a mitochondrial pyruvate carrier inhibitor used to assess metabolic dependence on pyruvate and consequently on glucose) to evaluate the dependency on energy substrates. For each panel, data are representative of four independent experiments (n = 4) and are indicated as means ± SD. Statistical analysis was performed with one-way ANOVA followed by Tukey’s multiple comparisons test. In panels A and B, *** and **** indicate *p* < 0.001 or 0.0001, respectively. In panels C and D, **, ***, and **** indicate *p* < 0.01, 0.001, or 0.0001, respectively, between untreated cells and cells treated with IGF-1 for 24 h.

**Figure 5 ijms-25-13640-f005:**
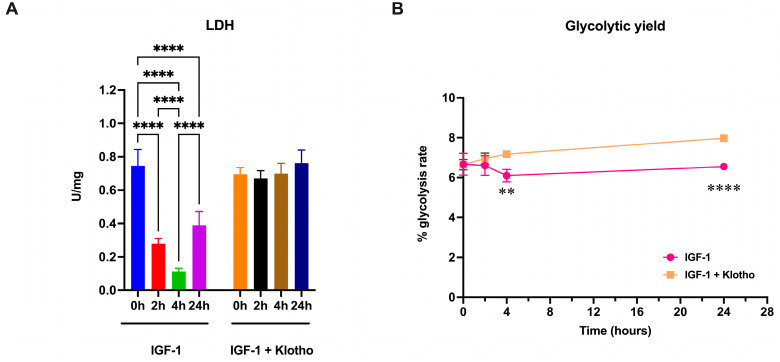
Anaerobic glycolytic metabolism in ARPE-19 cells treated with IGF-1 in the absence or presence of Klotho. All data were obtained from ARPE-19 cells treated with IGF-1 for 2, 4, and 24 h in the absence or presence of Klotho. (**A**) Lactate dehydrogenase (LDH) activity. (**B**) Glycolytic yield. Data are representative of four independent experiments (n = 4) and are indicated as means ± SD. For Panel A, statistical analysis was performed with one-way ANOVA followed by Tukey’s multiple comparisons test; for Panel B, statistical analysis was performed by a *t*-test for each time point. ** and **** indicate *p* < 0.01 or 0.0001, respectively.

**Figure 6 ijms-25-13640-f006:**
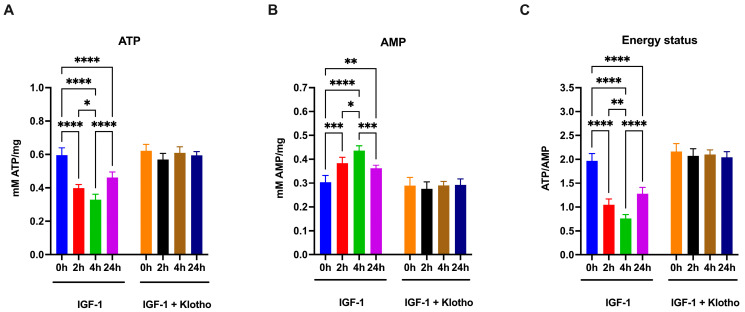
Cellular energy status of ARPE-19 cells treated with IGF-1 in the absence or presence of Klotho. All data were obtained from ARPE-19 cells treated with IGF-1 for 2, 4, and 24 h in the absence or presence of Klotho. (**A**) ATP intracellular concentration. (**B**) AMP intracellular concentration. (**C**) ATP/AMP ratio as a cellular energy status marker. Data are representative of four independent experiments (n = 4) and are indicated as means ± SD. Statistical analysis was performed with one-way ANOVA followed by Tukey’s multiple comparisons test. *, **, ***, and **** indicate *p* < 0.05, 0.01, 0.001, or 0.0001, respectively.

**Figure 7 ijms-25-13640-f007:**
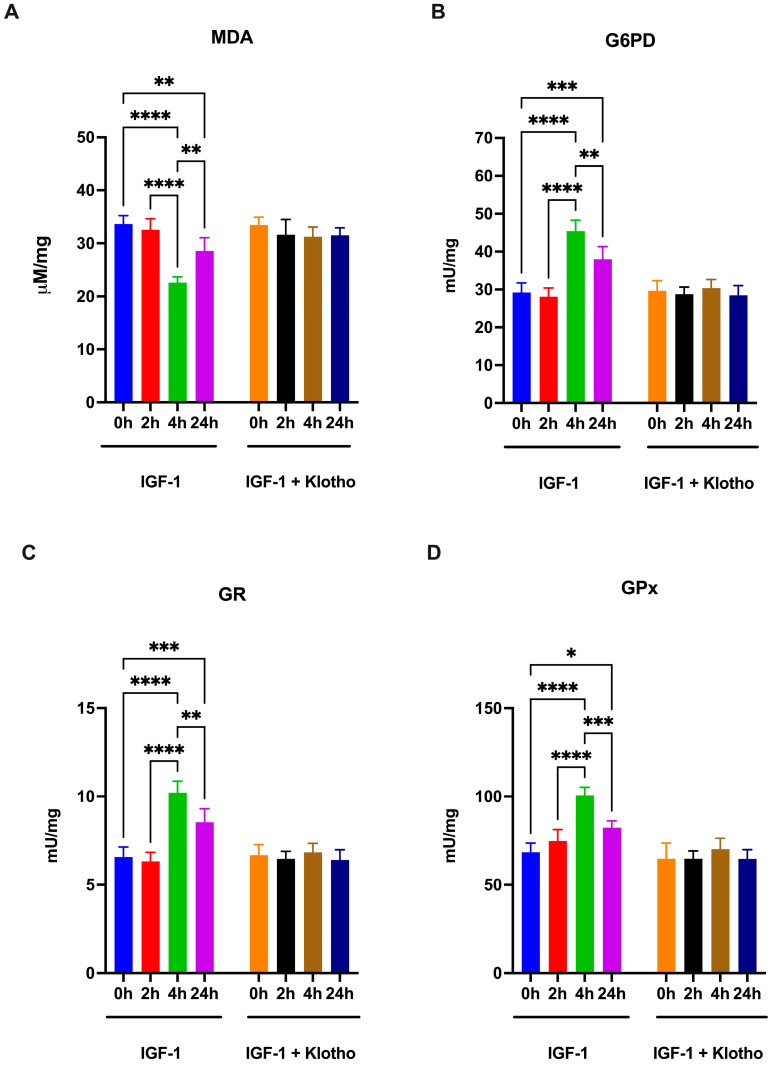
Lipid peroxidation accumulation and glucose-6-phosphate dehydrogenase (G6PD), glutathione reductase (GR), and glutathione peroxidase (GPx) activity in ARPE-19 cells treated with IGF-1 in the absence or presence of Klotho. All data were obtained from ARPE-19 cells treated with IGF-1 for 2, 4, and 24 h in the absence or presence of Klotho. (**A**) Malondialdehyde (MDA) intracellular concentration as a lipid peroxidation marker. (**B**) G6PD activity. (**C**) GR activity. (**D**) GPx activity. Data are representative of four independent experiments (n = 4) and are indicated as means ± SD. Statistical analysis was performed with one-way ANOVA followed by Tukey’s multiple comparisons test. *, **, ***, and **** indicate *p* < 0.05, 0.01, 0.001, or 0.0001, respectively.

**Figure 8 ijms-25-13640-f008:**
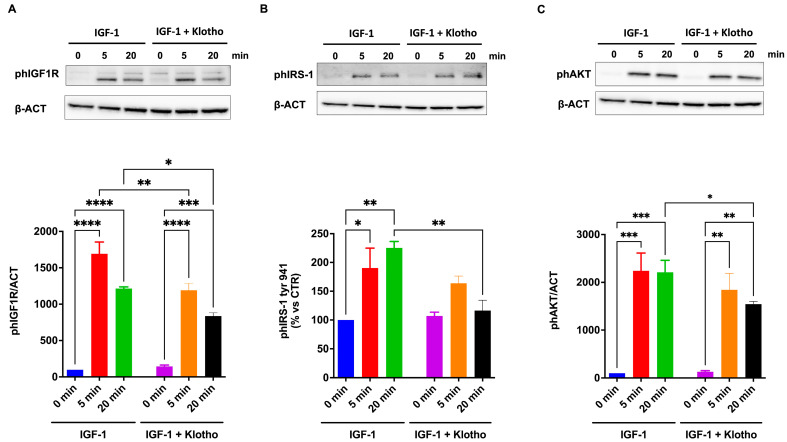
Intracellular signaling of IGF-1 in ARPE-19 cells. All data were obtained from ARPE-19 cells serum-starved overnight, then exposed to 100 nmol/L IGF-1 for 5 or 20 min in the absence or presence of Klotho. Then, cells were lysed and immunoblotted with anti-phospho antibodies against IGF1R (**A**), IRS-1 (**B**), and AKT (**C**). Representative Western blotting and the quantification of densitometries of Western blot bands are shown. Data are expressed as mean ± SD of fold induction relative to GAPDH (n = 3). Statistical analysis was performed with one-way ANOVA followed by Tukey’s multiple comparisons test. *, **, ***, and **** indicate *p* < 0.05, 0.01, 0.001, or 0.0001, respectively.

**Figure 9 ijms-25-13640-f009:**
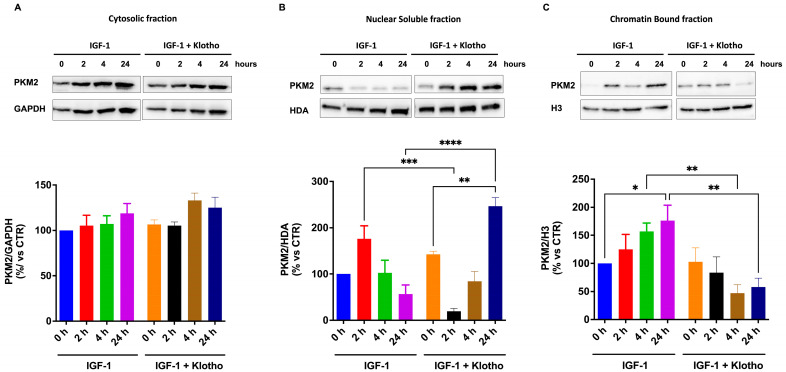
Subcellular localization of isoform M2 of pyruvate kinase (PKM2) in ARPE-19 cells treated with IGF-1 in the absence or presence of Klotho. ARPE-19 cells were treated with IGF-1 for 2, 4, and 24 h in the absence or presence of Klotho. Then, subcellular fractions of cells were obtained ((**A**) cytosol; (**B**) nuclear soluble; (**C**) chromatin bound). Representative Western blotting and quantification of the densitometries of Western blot bands are shown. Data are expressed as mean ± SD of fold induction relative to the amounts of each respective loading control (n = 3). Statistical analysis was performed with one-way ANOVA followed by Tukey’s multiple comparisons test. *, **, ***, and **** indicate *p* < 0.05, 0.01, 0.001, or 0.0001, respectively.

## Data Availability

All the data are contained within the article and the Appendix A.

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
