# Peer review of "IGF-1 Signaling Modulates Oxidative Metabolism and Stress Resistance in ARPE-19 Cells Through PKM2 Function"

_ijms, 2024, doi:10.3390/ijms252413640_

Round 1
Reviewer 1 Report
Comments and Suggestions for Authors
The manuscript entitled “IGF-1 Signaling Modulates Oxidative Metabolism and Stress Resistance in ARPE-19 Cells Through PKM2 Function” examines the impact of PKM2 modulation, mediated by IGF-1, on energy metabolism and redox balance in the retinal pigment epithelium ARPE-19 cell model. Additionally, the potential of Klotho protein as a means of preserving this process is investigated.
The manuscript is well written and easy to understand. The experiments carried out were enough and suitable for the purpose of the manuscript. The references used in the manuscript are recent and adequate. Regarding the novelty of the manuscript, as far as I am concerned, this is the first time that the IGF-1 modulation using Klotho protein in a retinal pigment epithelium model using ARPE-19 cells has been studied.
In my opinion, the results shown in the present manuscript are interesting for a broader community and deserve to be published, but the manuscript has some issues that need to be addressed:
Page 2, line 79: I would introduce the role of Klotho in line 75 rather than in line 79.
Use italics for statistic p.
Page 7 Line 146: Use superscript for the + symbol
Page 8 line 181: Define MDA, GR and GPx on its first appearance in the text.
Figure 7: There is no need to specify the nd when p is higher than 0.05; Acronyms used in the figure caption should be defined again as Figures are independent from the manuscript.
Page 12 line 272: There is no need to define PPP again, it was already defined in Page 8.
Page 13 line 319: When writing temperatures separate the number from the degree symbol to follow IUPAC recommendations. It is written both ways in the manuscript.
Page 13 line 319: Use subscript for 2 in CO2.
References do not follow the instructions for authors, as journal names should be abbreviated.
Best regards
Author Response
Please s
Reviewer 1
The manuscript entitled “IGF-1 Signaling Modulates Oxidative Metabolism and Stress Resistance in ARPE-19 Cells Through PKM2 Function” examines the impact of PKM2 modulation, mediated by IGF-1, on energy metabolism and redox balance in the retinal pigment epithelium ARPE-19 cell model. Additionally, the potential of Klotho protein as a means of preserving this process is investigated.
Comment 1: The manuscript is well written and easy to understand. The experiments carried out were enough and suitable for the purpose of the manuscript. The references used in the manuscript are recent and adequate. Regarding the novelty of the manuscript, as far as I am concerned, this is the first time that the IGF-1 modulation using Klotho protein in a retinal pigment epithelium model using ARPE-19 cells has been studied.
Response 1: We thank the reviewer for the positive comment and for appreciating the originality of the topic.
Comment 2: In my opinion, the results shown in the present manuscript are interesting for a broader community and deserve to be published, but the manuscript has some issues that need to be addressed:
Page 2, line 79: I would introduce the role of Klotho in line 75 rather than in line 79.
Use italics for statistic p.
Page 7 Line 146: Use superscript for the + symbol
Page 8 line 181: Define MDA, GR and GPx on its first appearance in the text.
Figure 7: There is no need to specify the nd when p is higher than 0.05; Acronyms used in the figure caption should be defined again as Figures are independent from the manuscript.
Page 12 line 272: There is no need to define PPP again, it was already defined in Page 8.
Page 13 line 319: When writing temperatures separate the number from the degree symbol to follow IUPAC recommendations. It is written both ways in the manuscript.
Page 13 line 319: Use subscript for 2 in CO2.
References do not follow the instructions for authors, as journal names should be abbreviated.
Response 2: We thank the reviewer for these suggestions and apologize for the mistakes. In the revised version, we have modified the text according to the reviewer’s recommendations.
ee the attachment.
Reviewer 2 Report
Comments and Suggestions for Authors
Dear authors,
It was a pleasure to review your manuscript titled "IGF-1 signaling modulates oxidative metabolism and stress resistance in ARPE-19 cells through PKM2 function". Your work addresses critical issues in the field of retinal metabolic dysfunction and its connection to degenerative diseases. Specifically, the focus on the role of PKM2 modulation by IGF-1 and the protective potential of Klotho is both timely and scientifically significant, providing valuable insights into the molecular mechanisms underlying retinal health and disease.
The manuscript is well-structured and contributes meaningfully to the understanding of how IGF-1 signaling impacts oxidative metabolism and stress resistance in ARPE-19 cells. The experimental approach is thorough, and the results shed light on key pathways that could have implications for therapeutic strategies targeting retinal degenerative diseases.
However, there are a few limitations that I believe should be addressed before publication to enhance the impact and clarity of the manuscript.
- Figure 1B. The expression of dimeric PKM2 at 24 hours does not appear as a clear band but rather as a broad point in the center. Meanwhile, the band behind it seems less intense compared to the tetramer or monomer. Could you please clarify what this signifies and how it might affect the quantification?
- Figure 2, the results displayed are not clear. To improve the quality and interpretability of this figure, I recommend using specific markers for the nucleus and cytoplasm (e.g., cytoskeletal markers) to evaluate colocalization effectively. Additionally, increasing the magnification would allow for a more detailed view of what you aim to demonstrate. It is also important to include a scale bar to ensure clarity and accuracy in the presentation of the images.
- Figure 3 does not specify the sample size (n) used in the experiments.
- In section 2.4, the exact rates of increase and decrease after 24 hours and 4 hours, respectively, are not specified. Could you include the exact values or provide a more detailed explanation of these dynamics?
- line 128, where it appears the reference should be to Figures 4C and 4D.
- Figure 8, the signal for IRS-1 in the Western blot appears suboptimal for accurate quantification, particularly given the small sample size of 3 per group. I suggest replacing this with a more representative and higher-quality blot.
- The statistical analysis section lacks sufficient detail, particularly regarding the protocol followed, which is critical for a quantitative study. Could you provide clarification on how the sample size (n) was calculated for the experiments? Additionally, there are instances, such as in Figure 1, where only three data points per group were used, yet an ANOVA was conducted. Could you explain how this was statistically valid and whether considerations for effect size and estimated statistical power were taken into account?
Author Response
Reviewer 2
Dear authors,
It was a pleasure to review your manuscript titled "IGF-1 signaling modulates oxidative metabolism and stress resistance in ARPE-19 cells through PKM2 function". Your work addresses critical issues in the field of retinal metabolic dysfunction and its connection to degenerative diseases. Specifically, the focus on the role of PKM2 modulation by IGF-1 and the protective potential of Klotho is both timely and scientifically significant, providing valuable insights into the molecular mechanisms underlying retinal health and disease.
Comment 1: The manuscript is well-structured and contributes meaningfully to the understanding of how IGF-1 signaling impacts oxidative metabolism and stress resistance in ARPE-19 cells. The experimental approach is thorough, and the results shed light on key pathways that could have implications for therapeutic strategies targeting retinal degenerative diseases.
Response 1: We thank the reviewer for the positive comment and for appreciating the originality of the topic.
However, there are a few limitations that I believe should be addressed before publication to enhance the impact and clarity of the manuscript.
Comment 2: Figure 1B. The expression of dimeric PKM2 at 24 hours does not appear as a clear band but rather as a broad point in the center. Meanwhile, the band behind it seems less intense compared to the tetramer or monomer. Could you please clarify what this signifies and how it might affect the quantification?
Response 2: We thank you for your comment. The point on the PKM2 band at 24 hours is an artifact caused by chemiluminescent development. The analysis software excludes the light from this point during band quantification; therefore, it does not affect the result. Indeed, the signals corresponding to the PKM2 dimers are less intense than those of the monomers and tetramers. However, the main finding is that dimers begin to form in the presence of IGF-1, leading to an increased ratio between the dimeric and tetrameric forms of PKM2. This increase has two consequences: (1) a progressive decrease in PKM2 activity, as shown in Figure 3, and (2) the nuclear localization of PKM2 (Figure 9C). Thus, these findings establish a link between IGF-1 signaling and RPE cell metabolism.
Comment 3: Figure 2, the results displayed are not clear. To improve the quality and interpretability of this figure, I recommend using specific markers for the nucleus and cytoplasm (e.g., cytoskeletal markers) to evaluate colocalization effectively. Additionally, increasing the magnification would allow for a more detailed view of what you aim to demonstrate. It is also important to include a scale bar to ensure clarity and accuracy in the presentation of the images.
Response 3: We apologize for the low quality of the images presented in Figure 2. In the revised version, we have improved the resolution of the fluorescent signal and included a scale bar. We appreciate the Reviewer’s suggestion to add nuclear and cytosolic markers to highlight PKM2 localization better. However, we believe that nuclear staining with DAPI is sufficient to distinguish the nuclear from the cytosolic component, as another cytosolic marker might make it harder to appreciate the PKM2 signal in the cytosol.
As suggested by the reviewer, we have included image magnifications of two fields for each condition to better appreciate the localization of PKM2 in nucleus.
Comment 4: Figure 3 does not specify the sample size (n) used in the experiments.
Response 4: Data reported in Figure 3 derives from four independent experiments. This information has been added in the revised version.
Comment 5: In section 2.4, the exact rates of increase and decrease after 24 hours and 4 hours, respectively, are not specified. Could you include the exact values or provide a more detailed explanation of these dynamics?
Response 5: We apologize for the lack of detail regarding the aerobic metabolism data. In the revised version, we have included the rates of increase and decrease after 24 and 4 hours of treatment, proposing an explanation for the dynamics of these changes.
Comment 6: line 128, where it appears the reference should be to Figures 4C and 4D.
Response 6: We thank the reviewer for the comment and apologize for the mistake. In the revised version, the reference to the panels in Figure 4 has been corrected.
Comment 7: Figure 8, the signal for IRS-1 in the Western blot appears suboptimal for accurate quantification, particularly given the small sample size of 3 per group. I suggest replacing this with a more representative and higher-quality blot.
Response 7: We apologize for the quality of the image, which is due to the detection limit of the primary antibody. Following the Reviewer's suggestion, we replaced the image of phIRS-1 with a better-quality one.
Comment 8: The statistical analysis section lacks sufficient detail, particularly regarding the protocol followed, which is critical for a quantitative study. Could you provide clarification on how the sample size (n) was calculated for the experiments? Additionally, there are instances, such as in Figure 1, where only three data points per group were used, yet an ANOVA was conducted. Could you explain how this was statistically valid and whether considerations for effect size and estimated statistical power were taken into account?
Response 8: We thank the reviewer for this comment. Indeed, we considered unnecessary to calculate the sample size because we worked with a cell line that is not subject to the high variability typical of human samples or animal models.
Given that we aimed to compare four conditions — control, treatment after 2 hours, treatment after 4 hours, and treatment after 24 hours — we considered one-way ANOVA to be the most appropriate statistical test. This choice was made because the t-test is limited to comparisons between two variables at a time.